# Method for Improving the Reliability of SRAM-Based PUF Using Convolution Operation

Ruihu Cao [1,2], Niansong Mei [1,2,*] and Qian Lian [1,2]

1     Shanghai Advanced Research Institute, Chinese Academy of Sciences, Shanghai 201210, China
2     University of Chinese Academy of Sciences, Beijing 100049, China
*     Correspondence: meins@sari.ac.cn; Tel.: +86-021-2032-5151

**Abstract:** This paper introduces a novel and efficient physical unclonable function (PUF) extraction method for SRAM. The proposed one-layer convolution scheme is based on a convolution operation, which significantly enhances the reliability of the PUF. To further reduce the hardware resources, a lightweight solution is presented based on a one-layer convolution scheme at the cost of a higher redundancy coefficient and a larger range for the inter-chip Hamming distance (HD). Both the above schemes only use certain hardware resources in the initial stage and the hardware resources are automatically released after PUF verification. The two schemes were verified using SRAM cells in three stm32f407 chips to output a 256-bit PUF response. The experimental results show that the one-layer convolution scheme required 8 KB SRAM, while the lightweight scheme used only 0.5 KB SRAM. The reliability of the one-layer convolution was found to be 100% when the redundancy coefficient was 0.08 and the inter-chip HD was 50.8073%. The reliability of the lightweight scheme was 100% and of the inter-chip HD was 50.195%.

**Keywords:** SRAM PUF; convolution; Hamming distance; Internet of Things (IoT) security





## 1. Introduction

The PUF method has attracted much attention in the IoT security field because of its uniqueness and unpredictability. A silicon-based PUF utilizes random differences in the integrated circuit manufacturing process to generate the unique identity information (ID) for the chip [1]. This can be used for secure key and device authentication. Because most IoT devices are equipped with SRAM, SRAM-based PUF represents a promising technology for practical application.

SRAM PUF employs a SRAM cell (constructed as two cross-coupled inverters). The state of the cell is determined by the random mismatches in the pair of inverters [2]. However, the output of the SRAM cell is sometimes unstable due to thermal noise, PVT variations, aging, and other factors [3]. Unstable SRAM cells can reduce the reliability of SRAM PUF. Table 1 shows the SRAM PUF stability test results for different products at $-40\,^{\circ}\mathrm{C}$, $20\,^{\circ}\mathrm{C}$ and $80\,^{\circ}\mathrm{C}$ [4]. On average, about 10% of SRAM cells were unstable, with a worst performance of 14.8% instability. Unstable cells represent the main challenge in SRAM PUF design.

Commonly used methods to eliminate unstable cells in SRAM PUF include new SRAM structures, manual intervention and use of software algorithms. In [5], a new structure for a high-precision amplifier circuit was used to compare the resistance values of two RRAM arrays, with the accurate PUF output generated using a resistance difference method. This RRAM PUF achieved a lower than $6 \times 10^{-6}$ bit error rate. In [6,7], two PMOS transistors in the SRAM cell were replaced with two NMOS transistors to produce an enhancement-enhancement SRAM (EE SRAM), the observed reliability improving to 99.79%. The new SRAM structure was able to improve incidence of SRAM PUF stable cells; however, ordinary SRAM is usually used in existing IoT equipment, limiting the application of this technology.

**Table 1.** Instability rate of SRAM devices for different temperatures.

| SRAM | Technology | Devices | $-40\,°C$ | $+20\,°C$ | $+80\,°C$ |
|---|---|---|---|---|---|
| Cypress CY7C15632KV18 | 65 nm | 10 | 7.80% | 3.80% | 6.60% |
| Virage HP ASAP SP ULP 32-bit | 90 nm | 34 | 14.80% | 2.90% | 6.50% |
| Faraday SHGD130-1760X8X1BM1 | 130 nm | 40 | 10.30% | 4.50% | 9.00% |
| Virage asdsrsnfslp1750x8cml6swO | 130 nm | 40 | 12.00% | 4.80% | 10.50% |
| Cypress cY7C1041CV33-20Zsx | 150 nm | 8 | 6.70% | 3.50% | 8.00% |
| IDT 71V416S15PHI | 180 nm | 8 | 6.00% | 2.80% | 8.40% |

Manual intervention is a direct method. In [8], a negative bias temperature instability (NBTI) method was used to increase the mismatch of the inverter in SRAM, thus improving the stability of the SRAM PUF output. When the mismatch reached 50mV, the reliability was improved by more than 20%. Another direct method involves the detection and marking of unstable or potentially unstable locations [9,10]. The selected cells are temporarily 100% stable. The disadvantage of this scheme is that it takes a lot of time. In addition, initially stable cells can become unstable as the operating time increases.

To address the problems associated with the two methods described above, software algorithms have been introduced. Traditional software algorithms have used an error correction code to correct the response. The error correction code needs auxiliary data to recover data [11]. If the auxiliary data is damaged, the error correction code becomes invalid. However, it is difficult to deploy the fuzzy extractor in resource-constrained systems since the implementation of error correction codes is complicated. Error correction requires considerable hardware resources [12]. In [13,14], a convolutional neural network (CNN) model was used to classify the results of SRAM PUF to improve the stability of SRAM PUF. The reliability was improved to 97.34% [15]. However, the CNN model is only able to classify SRAM groups—it cannot create the ID number. Moreover, a digital image requires 256KB SRAM cells. The CNN model relies on the GPU to provide massive initial data [13]. The existing convolution model cannot be implemented on IoT devices.

A novel method to obtain a reliable SRAM PUF response is proposed in this paper. This method is applicable to all SRAM circuits and does not consume additional hardware resources. In this scheme, a software algorithm is used to extract PUF with high reliability. In the process of PUF extraction, the SRAM only needs to provide the initial value. After PUF extraction, the SRAM cells can still be used as normal SRAM cells. The following sections describe the scheme in detail. Section 2 describes the basis of convolution. Section 3 introduces the workflow for the scheme and proposes a lightweight scheme for IoT devices. Section 4 verifies the feasibility and reliability of the schemes. Section 5 provides a summary and conclusions.

## 2. Related Work

Convolution filtering is used to process random noise in digital-image processing. Ambient noise is received during image acquisition. The noise forms fuzzy points in a two-dimensional matrix. Convolution can attenuate the effect of random noise, such as salt and pepper noise and Gaussian white noise. Convolution uses the convolution kernel to extract information. As shown in Figure 1, initial images can be filtered by convolution kernels in two main ways. Convolution kernels and strides adjust the size of the final results.

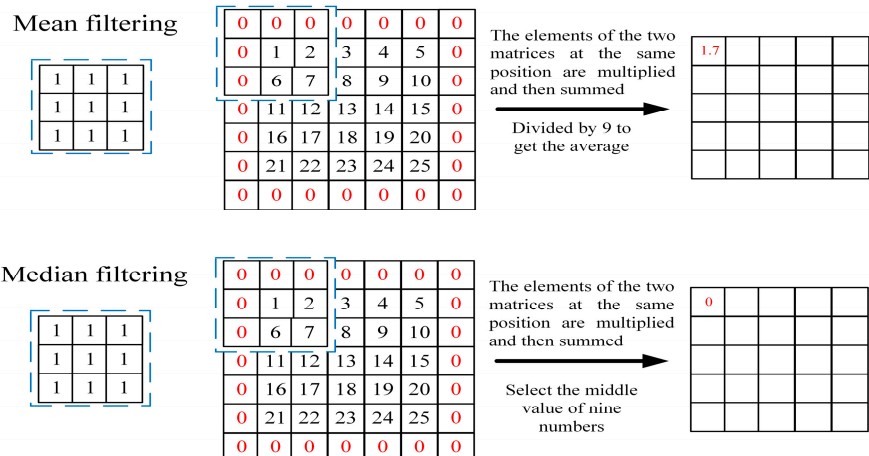

**Figure 1.** Difference between mean filtering and median filtering.

The signal consists of the true value and noise. In a $3 \times 3$ flat area, the true values are f0 and the noise satisfies a Gaussian distribution. The signal can be expressed as:

$$g = f + n \tag{1}$$

where f is the actual value and n is the additive Gaussian noise. After mean filtering:

$$g' = f0 + (n0 + n1 + \ldots + n8)/9 = f0 + n' \tag{2}$$

The average of the true value is f0, and the average of the noise is n'. The influence of the noise is attenuated by mean filtering. Mean filtering entails a linear filter that does not protect image details. Median filtering selects the middle value of the numbers in the convolution kernel. Median filtering is a non-linear filter that protects the sharp edges of the image.

This paper assumes that the initial power-on value of the SRAM is a two-dimensional matrix composed of 0 and 1. The unstable SRAM cell is like a noise element in the digital image. The unstable cells in SRAM PUF are caused by both internal noise and external noise. The position affected by the noise is random.

### 3. The Proposed Scheme

#### 3.1. Common Convolution Filtering

Common convolution uses filtering to reduce noise in digital-image processing, such as by using mean filtering and median filtering. For example, it can filter the large matrix to a small matrix by one or two layers of convolution through the equal-stride convolution kernel. The step stride of the convolution is the same as the length of the convolution kernel. The mean filtering and median filtering approaches were tested for our SRAM data to determine the feasibility of common convolution. However, the output results were unstable. The common convolution instability rate was higher than that for the original SRAM.

The outputs of digital-image processing are 8-bit pixels. In contrast, SRAM PUF binarizes the value of the output. A value greater than 0.5 is regarded as 1, and a value less than or equal to 0.5 is regarded as 0. Convolution does not remove noise—it only reduces the weight of the noise in pictures. The mean value of an area tends to be 0.5 for a matrix which has equal probability random distributions of 0 and 1. Unstable cells cause the mean value of an area to fluctuate around 0.5. If the noise changes the results of binarization, the convolution output becomes unstable.

Convolution filtering avoids the issue of the need for additional hardware resources associated with traditional SRAM PUF. All the computing work can be undertaken by the server side of the verifier. When receiving the verification instruction, the MCU system only needs to provide the SRAM power-on initial value for a certain area. When the authentication is completed, this area can still undertake other functions of MCU, as with normal SRAMs. Convolution filtering represents a novel way to generate SRAM PUF. This paper presents a new scheme to improve the reliability of convolution.

*3.2. One-Layer Convolution Scheme*

The proposed scheme includes a standard matrix and a verification matrix. The standard matrix is obtained by calculating the average value of 100-times the initial power-on value and binarizing it. The output result is determined by the standard matrix. The PUF can better discriminate using the standard matrix generated by the average value. The initial value for any time can also be adopted for the standard matrix. Different standard matrices will produce different results. These approaches do not affect the reliability of the proposed scheme.

The one-layer convolution scheme generates an $m \times m$ standard matrix at the SRAM PUF registration stage. The convolution process calculates the sum of the elements in the standard matrix through an $n \times n$ convolution kernel. The step size of the convolution is the same as the length of the convolution kernel. The new matrix generated is called the verification matrix, the size of which is $(m/n) \times (m/n)$ ($m$ can be divided by $n$). The verification matrix, standard matrix and the address of SRAM need to be entered into the validator's database. The scheme sets a redundancy coefficient. The redundancy coefficient is a value that is determined according to different types of SRAM. The probabilities for the cells from 0 to 1 and from 1 to 0 are the same. The formula to calculate the redundancy coefficient is as follows.

$$\frac{N}{2} * \frac{L}{S} = \text{Redundancy coefficient} \tag{3}$$

where N is the naturally unstable rate of SRAM, L is the length of the convolution kernel and S is the step size of the convolution. The redundancy coefficient multiplied by the size of the convolution kernel represents the redundancy. The redundancy indicates the maximum number of unstable cells in the range of the convolution kernel size. A small redundancy coefficient can effectively distinguish two similar SRAM datasets. However, if the noise causes the number of unstable cells to be larger than the redundancy, the output results will be unstable. A large redundancy coefficient indicates strong anti-interference ability, which implies better reliability. However, a large redundancy coefficient reduces the discrimination between two similar SRAM datasets. The difference between the maximum inter-chip HD and the minimum inter-chip HD will also be large. Therefore, it is necessary to set up a suitable redundancy coefficient to meet different performance requirements.

When an SRAM is authenticating, the scheme uses the same $n \times n$ convolution kernel for the new $m \times m$ data. It generates an unverified matrix for identity authentication. If the difference in the absolute value between the unverified matrix and the verification matrix in the same position is less than the redundancy, this position is stable. If the value of the stable positions in the verification matrix is larger than half of the size of the convolution kernel, the output signal is 1, otherwise it is 0. If it is an unstable position, the validator will output 0.5, indicating an abnormality. The workflow is shown in Figure 2.

During initialization, the processor verifies the standard matrix with the standard matrix to obtain the standard output of SRAM PUF and adds it into the validator's database.

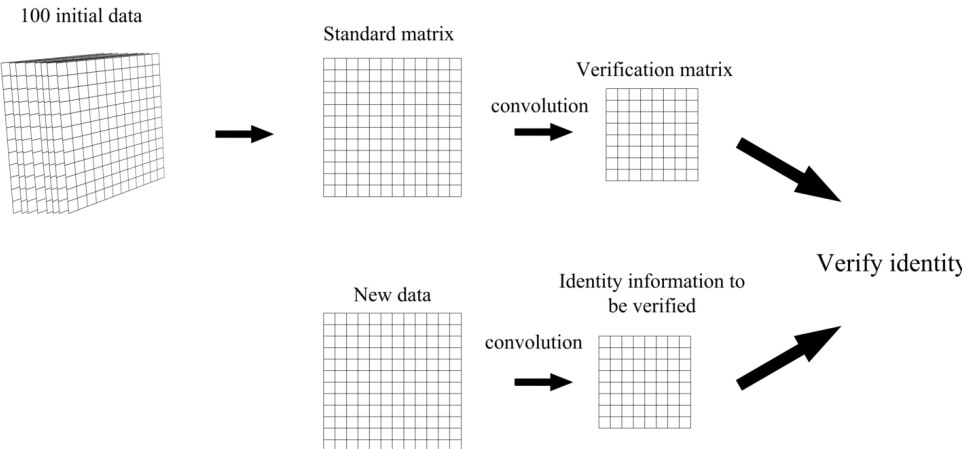

**Figure 2.** Workflow chart.

### 3.3. The Lightweight Scheme

Many IoT devices do not have a large number of SRAM cells. It is better to generate ID numbers of the same length with fewer SRAM cells, so we propose a lightweight scheme. The standard matrix is filled by self-filling and the step size of the convolution is half of the length of the convolution kernel. In the lightweight scheme, the standard matrix is $a \times a$, the convolution kernel is $b \times b$ and the length of self-filling is equal to the step size of the convolution. The size of the output result is $(\frac{a}{b/2}) \times (\frac{a}{b/2})$. If $a = m/4$ and $b = n/2$, the lightweight scheme uses one-sixteenth of the hardware resources above to generate the same length ID number.

Because there may be some unstable cells that are repeatedly calculated, the convolution is computed with a step size of $b/2$. The redundancy coefficient of the lightweight scheme is larger than the one-layer convolution scheme, according to Formula 3. A large redundancy coefficient can ensure 100% reliability of the lightweight scheme. However, the discrimination ability and inter-chip HD of the lightweight scheme may not be as good as for the one-layer convolution scheme. Since cells with initial values of 0 and 1 have the same probability of being affected by aging, use of the convolution operation can offset the effect of aging on reliability.

### 4. Implementation Results

The verification process for the schemes used the CCM area of the STM32F407 chip shown in Figure 3. The 64 Kbyte CCM (core coupled memory) data RAM was a 64 KB SRAM dedicated to the F4 core. The CCM was not part of the bus matrix, and these SRAMs were not given a fixed initial value. The external device was only able to read the initial value but was not able to change it. This precisely fitted the needs of SRAM PUF. Because 1 byte is equal to 8 bits in the MCU, the $a$, $b$, $m$, $n$ values should be integer multiples of 8. The schemes generated a 256-bit ID number in the experiments described below.

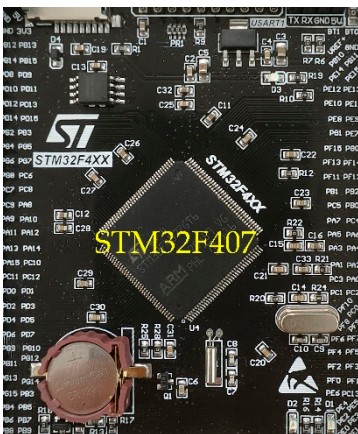

**Figure 3.** The STM32F407 used in the experiment.

### 4.1. The Results of One-Layer Convolution Scheme

The one-layer convolution scheme uses a 256 × 256 standard matrix. The convolution calculates the sum of the elements in the standard matrix through a 16 × 16 convolution kernel. The step size of the convolution is 16. The verification matrix generated is a 16 × 16 matrix. Our test data show that about 10% of the SRAM in the CCM area of this chip was unstable. So, the redundancy coefficient was first set to 0.05 according to Formula 3, meaning that the redundancy was 0.05 times the convolution kernel size.

Using the proposed scheme, the 256 × 256 matrix data was convoluted to obtain 16 × 16 response information. The absolute address was 0X10000000. The results showed that the 5, 7, 24, 26, 45, 53, 55, 63, 67, 70, 76, 83, 97 groups had errors in their own data. The group with the most errors was the 83rd group, which had four error positions, as shown in Figure 4. The error numbers of the other groups were all one or two. The total number of errors was 26. It should be emphasized that the outputs of the 100 groups had a total of 16 × 16 × 100 bits. The reliability of this scheme was 99.8984% when the redundancy coefficient was 0.05.

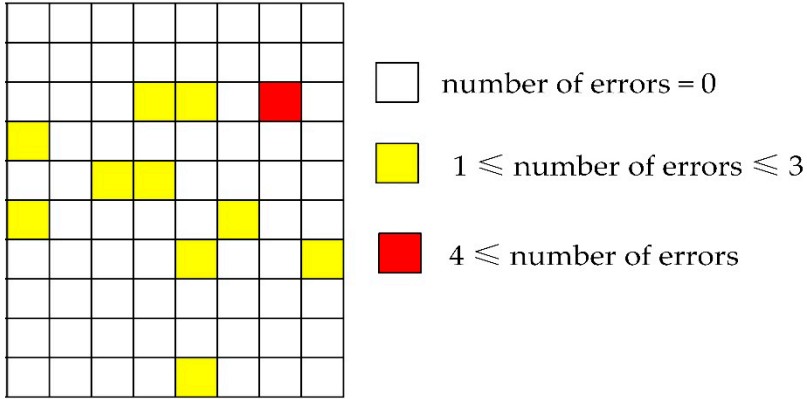

**Figure 4.** Location and number of errors in one verification.

The data for the absolute address 0X10005700 had 29 total errors, with a reliability rate of 99.8867%. The absolute address 0X10007000 had 77 total errors, with a reliability rate of 99.6999%. The absolute address 0X1000ABC0 had 16 total errors, with a reliability rate of 99.9375%. The absolute address 0X1000B000 had 5 total errors, with a reliability rate of 99.9805%. The absolute addresses, total number of errors and reliabilities are shown in Table 2.

**Table 2.** Total number of errors and reliability of the five groups (redundancy coefficient = 0.05).

| Absolute Address | Total Number of Errors | Reliability (%) |
|---|---|---|
| 0X10000000 | 26 | 99.8984 |
| 0X10005700 | 29 | 99.8867 |
| 0X10007000 | 77 | 99.6999 |
| 0X1000ABC0 | 16 | 99.9375 |
| 0X1000B000 | 5 | 99.9805 |

A small adjustment to the redundancy coefficient can improve the reliability of the scheme. The redundancy coefficient was changed to 0.06 and the verification process was performed for the above data again. The results were as follows: For the absolute address 0x10005700, the total error was 2, and the reliability was 99.992%. For the absolute address 0x10007000, the total error was 6, and the reliability was 99.97656%. The reliabilities for all other areas were 100%. When the redundancy coefficient was increased to 0.08, all reliabilities were able to reach 100%. Before initializing the SRAM PUF, it is necessary to adjust the redundancy coefficient for high reliability according to the different types of SRAM.

The standard result for 0X10000000 was compared with the standard result for 0X10005700 bit-by-bit. A total of 45.7% of the elements in the same position had different values. Ten groups of areas were selected with absolute addresses 0X10000000, 0X10001000, 0X10002000, 0X10003500, 0X10005700, 0X10007000, 0X10009000, 0X1000ABC0, 0X1000B000, 0X10004A00. The inter-chip HD is shown in Figure 5. The average inter-chip HD for all data was 50.8073%. The standard outputs generated were around 50%, which means that the scheme exhibited good uniqueness.

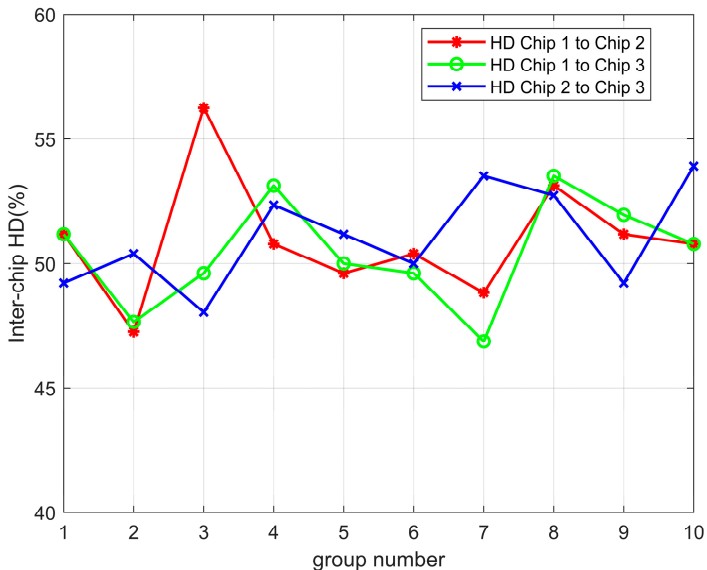

**Figure 5.** The inter-chip HD in one-layer convolution.

### 4.2. The Results of Lightweight Scheme

The lightweight scheme is suitable for IoT devices. Stable 256-bit information can be obtained from $64 \times 64$ bit initial data using a new convolution. The size of the convolution kernel is $8 \times 8$. The step size of this convolution is four bits, as shown in Figure 6. This lightweight scheme uses 64 bits SRAM to output a one-bit response. The hardware cost is half of the traditional Bose–Chaudhuri–Hocquenghem (BCH) codes scheme.

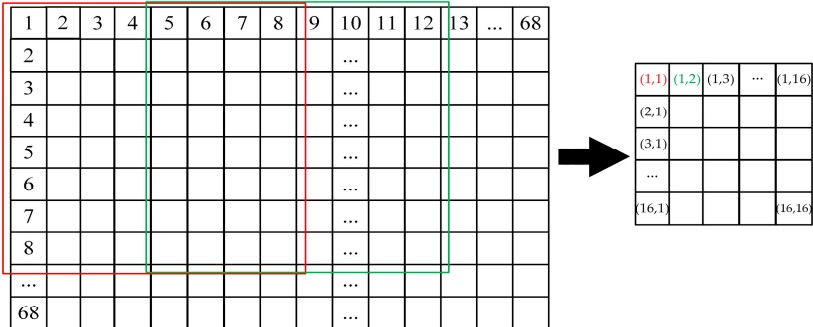

**Figure 6.** Convolution operation with self-filling.

Ten groups of areas were selected with absolute addresses 0X10000000, 0X10001000, 0X10002000, 0X10003500, 0X10005700, 0X10007000, 0X10009000, 0X1000ABC0, 0X1000B000, 0X10004A00. The redundancy coefficient was changed to 0.1 according to Formula 3. In Figure 7, a comparison of the inter-chip HD of three chips is presented. The results were all around 50%. The average inter-chip HD for all data was 50.195%, which indicates that the results of this scheme are unique. The result showed a large swing around 50%, which confirms the analysis provided in Section 3.3. The reliabilities of the three chips were 100% for all the above addresses.

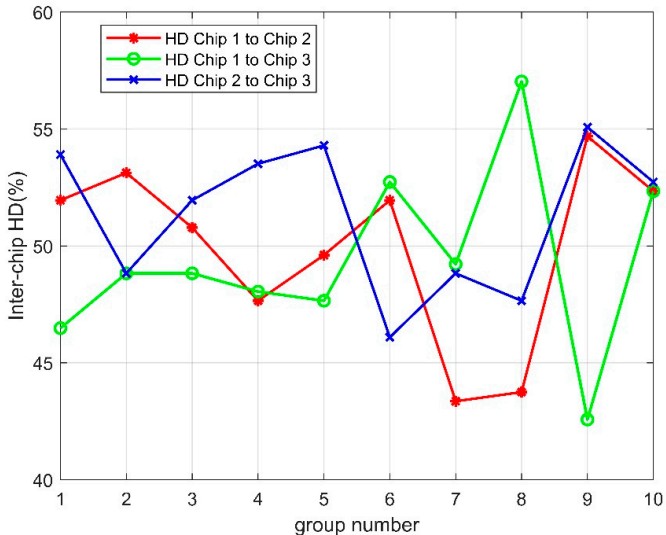

**Figure 7.** The inter-chip HD in the lightweight scheme.

Different temperatures have a large impact on the stability of SRAM, so the initial SRAM values for three chips in the same region were tested at temperatures of $-20$ °C, 40 °C and 80 °C. The data at 20 °C was used as the standard matrix and verification matrix. The algorithm above was used to process the same location data at different temperatures. The SRAM worked normally as it would at 20 °C at $-40$ °C and 80 °C. As is shown in Table 3, the three chips produced the same results at different temperatures based on the verification matrices that were saved, indicating that the lightweight scheme was able to perform well at different temperatures.

**Table 3.** The performance of absolute address 0X10004A00 in three chips.

| The Absolute Address 0X10004A00 | Output Result (HEX) | BER at $-40\,^\circ$C | BER at $80\,^\circ$C | Inter-Chip HD |
|---|---|---|---|---|
| No. 1 chip (20 $^\circ$C) | e3c1 f38f 03be 07fc 3ef4 3937 fd1e ec07 fb16 489e 083e 9f66 8fc6 8676 b677 9fe0 | 0 | 0 | 51.56% (No. 1 to No. 2) |
| No. 2 chip (20 $^\circ$C) | 1fcf c7f7 e113 641f 06e2 8ce0 fce1 fdc1 3cc1 ffcc 73fc 160f 9346 7101 3013 11c3 | 0 | 0 | 50.78% (No. 2 to No. 3) |
| No. 3 chip (20 $^\circ$C) | 7c19 7539 23e1 8360 6e6f f281 9380 8fe2 cfe1 e0e2 9283 1837 3be0 7f81 0798 9be0 | 0 | 0 | 52.34% (No. 1 to No. 3) |

*4.3. Comparison with Other Schemes*

Table 4 compares results for the scheme and previously published reliability data for SRAM PUFs.

**Table 4.** Performance and comparison with prior studies.

| | This Work | JUNE [6] | TCAS I [16] | Access [17] | ISSCC [18] | ISSCC [19] |
|---|---|---|---|---|---|---|
| Technology (nm) | 90 | 130 | 65 | 65 | 65 | 28 |
| Topology | SRAM | EE SRAM | SRAM | SRAM | EE SRAM | SRAM |
| Stabilizing technique | Convolution | Vss bias dark-bit detection | Capacitive digital preselection | Capacitive tilt, mirror | Automatic self-checking and healing | SMV, BCH |
| Native unstable cells | 10% | 2.14% | 19.6% | 42.1% | 2.8% | 10.5% |
| Stabilized BER | 0 [a] | 0 [b] | $<1.4 \times 10^{-9}$ | $2.6 \times 10^{-6}$ | $<3.3 \times 10^{-8}$ | $8.9 \times 10^{-4}$ |
| Measure temp. ($^\circ$C) | $-40$–80 [c] | $-40$–150 | $-10$–80 | $-10$–85 | $-40$–125 | $-40$–150 |
| %1's in the qualified cells | 52.85% | – | 51.5% | 50.1% | – | 48.5% |
| Inter-chip HD | 50.5% [d] | – | 49.9% | 49.3% | – | 49.8% |

[a]. No error in 256 bits, with 30,000 evaluations in 3 different chips in lightweight scheme; redundancy coefficient is 0.08 in one-layer convolution scheme. [b]. No error in 3339 bits with 500 evaluations. [c]. Test for the lightweight scheme. [d]. For all results in this paper.

The proposed scheme can provide response information with 100% reliability using a convolution algorithm with redundancy. It does not require additional hardware resources and can adjust flexibly according to different needs.

**5. Conclusions**

This paper introduces a novel method consisting of a one-layer convolution scheme and a lightweight scheme for SRAM PUF that improves the reliability and hardware efficiency. The novel convolution generates standard and verification matrices during initialization. The verification matrix is obtained from the standard matrix and the PUF response is generated by comparing the verification matrix and unverified matrix values. Moreover, the lightweight scheme uses one-sixteenth of the one-layer convolution scheme hardware resources to generate the same length results; however, it has a larger redundancy coefficient, which reduces its discrimination and the inter-chip HD range. The results of the three chips obtained at 30,000 data sets demonstrate the proposed scheme's performance in terms of 256-bit response outputs. The reliability of all responses was shown to be able to reach 100%, with an inter-chip HD of around 50%. Moreover, the hardware cost per bit for the light-weight scheme was half that for traditional BCH, greatly reducing resource usage. The SRAMs used in the novel convolution method can work like normal SRAMs following SRAM PUF; therefore, the method can be combined with lightweight IoT devices under different conditions. In addition, the identity verification process requires only the initial value of the SRAM to be processed on the verifier's server and does not need MCU resources. The aging problem can, therefore, be avoided. Further development of the scheme represents a promising direction for future research.

**Author Contributions:** Conceptualization, R.C. and N.M.; methodology, R.C.; software, R.C.; validation, R.C.; formal analysis, R.C.; investigation, R.C.; resources, R.C.; data curation, R.C.; writing—

original draft preparation, R.C.; writing—review and editing, N.M. and Q.L.; visualization, R.C.; supervision, N.M.; project administration, N.M.; All authors have read and agreed to the published version of the manuscript.

**Funding:** This research received no external funding.

**Conflicts of Interest:** The authors declare no conflict of interest.

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
