# Peer review of "Method for Improving the Reliability of SRAM-Based PUF Using Convolution Operation"

_electronics, doi:10.3390/electronics11213493_

Round 1

Reviewer 1 Report

This paper proposes a convolution scheme for SRAM PUF to improve reliability and hardware efficiency. It generates a verification matrix using the SRAM bank. A validator compares the verification matrix with an unverified input data matrix to generate a response.

Pros: The proposed scheme shows a reliability of 100% and Inter-chip Hamming distance of 50% for 30000 sets of data on 3 chips. 

Cons: 

  1. Although the authors mention the value for the redundancy coefficient to be 5% of the validation value in the Implementation Section (Section. 4), it is unclear how the redundancy coefficient affects the output. What will happen if the response value exceeds 5% of the validation value?

  2. Please check for typographical errors in the manuscript. For example, Line 68, page 2; line 183, page 6.

  3. The implementation section uses validation value, which is nowhere mentioned in the document. What is the validation value that is being referred to in line 174, page 6? It is unclear if the verification matrix and validation values mean the same.

  4. The authors mention that they test for the chips at different temperatures but haven’t identified them in any of the results. A table/plot for the inter-chip HD and reliability for different temperatures would be helpful.

Author Response

Thank you for your  help and patience. We have gave answers to your suggestion. Please see the attachment.

Reviewer 2 Report

Recommendation: Reject; it appears that publication in any form would be premature at this time.

The main concerns are:

a)       Paper presentation.

§  The publication has been written in a very poor English, in several sections the text is incomprehensible thus the reader has to extrapolate somehow the useful information from a set of words.

§  The quality of the figures is awful: in figure 1 the median filtering is presented which, a part of a brief remark, is never used. In figure 2 there is “idendidy” instead of “identity”, or so I suppose. Figure 4 is taken from the stm32f407 datasheet which has never been cited in the text. Figure 5 can be substituted by a color map. Figure 7 is completely out of scale. And that’s more than a half of the content in the presented paper.

b)      Contents. The authors present an algorithm to reduce the instability of the SRAM cells used in the context of the PUF. Below some critical issues are listed:

§  In the introduction authors list different methods to reduce the SRAM instability stating that “A digital image requires 256KB SRAM cells”. Why is that? Why a digital image can not require less space? If the authors are referring to Najafi, F. , et al cited above, it uses 220KB SRAM cells. Also author states that it is necessary a GPU to classify a digital image, but again in Najafi, F. they use a simple FPGA. If it has nothing to do with the mentioned reference it is all the same, I can’t imagine why an entire GPU is required to implement a convolution on a 256x256 bit image.

§  Authors propose to post-process the data read from the SRAM. It is clear that by averaging and filtering any kind of noisiness can be suppressed. Besides that, authors propose to implement the post processing on the server side. In this case the key question is: so what is the novelty here in the context of the IoT?

§  In section 3 the proposed scheme should be generalized to a random size. The presentation of the proposed method is confusing: in sec.3.1 the filtering method is presented for a 256x256 matrix, while in the next section the lightweight scheme is proposed for a 64x64 matrix stating that usually IoT devices do not contain a large amount of SRAM cells. Furthermore there is a huge inconsistency between the text in section 3.2 and Figure. 3: in the text the kernel is 8x8 transforms in a 5x5 one in the figure, even worse, according to the figure 3 the convolution between 64x64 matrix 5x5 kernel results in a 5x5 matrix, which is totally wrong. But then, why propose 256x256 scheme if it is not usually the case according to the authors? Maybe it is a misunderstood due to a terrible text quality, but I believe that a more generic procedure should be developed instead of a bunch of application cases. Or  should the server side somehow recognize in which particular case, I may only guess.

§  The state-of-art comparison shows a competitive performance of the proposed work, however the comparison is not correct as the competitors actually deal with the IoT side of the PUF.

Author Response

(The authors gave the same response as above.)

Round 2

Reviewer 1 Report

The authors incorporated all my previous concerns in the manuscript.

Reviewer 2 Report

Recommendation: Accepted.

The submitted paper has been heavily modified improving the quality of the text and changing some figures by taking into account the suggestions from the previous review. In response letter authors gave detailed answers concerning the contents of the submitted work. Said that, I recommend the publication of this paper.